# Use of the Natural Outdoor Environment in Different Populations in Europe in Relation to Access: Implications for Policy

**DOI:** 10.3390/ijerph19042226

**Published:** 2022-02-16

**Authors:** Daniel Masterson, Margarita Triguero-Mas, Sandra Marquez, Wilma Zijlema, David Martinez, Christopher Gidlow, Graham Smith, Gemma Hurst, Marta Cirach, Regina Grazuleviciene, Magdalena Van den Berg, Hanneke Kruize, Jolanda Maas, Mark Nieuwenhuijsen

**Affiliations:** 1Centre for Health and Development, Staffordshire University, Leek Road, Stoke-on-Trent ST4 2DF, UK; C.Gidlow@staffs.ac.uk (C.G.); grahamsmith193@gmail.com (G.S.); G.L.Hurst@staffs.ac.uk (G.H.); 2Jönköping Academy for Improvement of Health and Welfare, Jönköping University, P.O. Box 1026, 551 11 Jönköping, Sweden; 3Mariana Arcaya’s Research Lab, Department of Urban Studies and Planning, Massachusetts Institute of Technology, Cambridge, MA 02139, USA; mtrigueromas@gmail.com; 4Barcelona Lab for Urban Environmental Justice and Sustainability, Institute for Environmental Science and Technology, Universitat Autònoma de Barcelona, 08193 Barcelona, Spain; 5IMIM (Hospital del Mar Medical Research Institute), 08003 Barcelona, Spain; 6Centre for Research in Environmental Epidemiology (CREAL), ISGlobal, Carrer del Dr. Aiguader, 88, 08003 Barcelona, Spain; sandra.marquez@isglobal.org (S.M.); wilma.zijlema@isglobal.org (W.Z.); mardav92@gmail.com (D.M.); marta.cirach@isglobal.org (M.C.); mark.nieuwenhuijsen@isglobal.org (M.N.); 7Department of Medicine and Life Sciences, Universitat Pompeu Fabra (UPF), 08002 Barcelona, Spain; 8CIBER Epidemiología y Salud Pública (CIBERESP), 08003 Barcelona, Spain; 9Department of Environmental Science, Vytauto Didžiojo Universitetas, 44248 Kaunas, Lithuania; r.grazuleviciene@gmf.vdu.lt; 10Department of Public and Occupational Health, Institute for Health and Care Research, VU University Medical Centre (VUMC), 1007 MB Amsterdam, The Netherlands; Magdalena.van.den.Berg@vggm.nl (M.V.d.B.); jolandamaas@hotmail.com (J.M.); 11National Institute for Public Health and the Environment, 3720 BA Bilthoven, The Netherlands; hanneke.kruize@rivm.nl

**Keywords:** natural outdoor environments, green space, access, use, distance, time spent

## Abstract

This cross-cultural study explores the relationship of natural outdoor environment (NOE) use with NOE access. Most urban planning recommendations suggest optimal accessibility to be 300 m–500 m straight distance to spaces with vegetation of at least 1 hectare. Exploring this recommendation, we used data (*n* = 3947) from four European cities collected in the framework of the PHENOTYPE study: Barcelona (Spain), Doetinchem (The Netherlands), Kaunas (Lithuania) and Stoke-on-Trent (United Kingdom) to obtain residential access to NOE (straight or network distances, using 300 m and 150 m buffers, to NOE larger than 1 hectare or 0.5 hectare) and use of NOE (i.e., self-reported time spent in NOE). Poisson regression models were used to examine the associations between residential access and use of NOE. The models with the strongest association with time spent in NOE in the combined sample were for those living within 300 m straight line distance to either 0.5 ha or 1 ha NOE. Noting that the only indicator that was consistent across all individual cities was living with 150 m network buffer of NOE (of at least 1 ha), this warrants further exploration in reducing recommendations of 300 m straight-line distance to 150 m network distance to 1 ha of NOE for a general indicator for cities within Europe.

## 1. Introduction

### 1.1. Natural Outdoor Environments and Health

The physical and mental health benefits of contact with natural outdoor environments (NOE) have long been appreciated and demonstrated by a growing number of studies [1,2,3,4]. NOE, also known as green and blue environments or green and blue spaces, are places with natural elements that can include vegetation and water bodies such as parks, canals, forests and recreation areas [5]. Recent studies show that proximity to and visiting NOE is associated with improved cardiovascular health [6,7,8]; reduced stress [9,10]; physical activity [5,11,12] and the amelioration of a range of socioeconomic and health-inequality-related issues [13,14]. Engaging with NOE can also provide opportunities for social interaction and promote community cohesion [15,16,17] and enhance vitality and mood [16,18,19,20,21]. It is beyond the scope of this paper to explore the underlying mechanisms of these wide-ranging benefits, though readers are directed to Bratman, Olvera-Alvarez and Gross [22] who explore theories such as the biophilia hypothesis and attention restoration theory. 

### 1.2. Access and Use of Natural Outdoor Environments (NOE)

Despite these wide-ranging benefits and understanding of the importance of the role urban design plays in promoting access [23], there remain considerable challenges in protecting existing NOE and ensuring adequate access to NOE in new urban developments. Access to NOE is increasingly considered an issue of environmental justice [21]. One such challenge is to establish optimal access to NOE (which includes distance to and size of NOE) to maximise use. Once these access standards are determined, they can be incorporated into planning policies and used in (re-)development designs. Without such guidance, access to NOE may be loosely defined and inappropriately applied or not adequately considered. Following a commitment to provide each child with access to NOE to play and partake in physical activity [24] and a subsequent review of evidence with proposed indicators for distance to NOE [25], a number of studies have explored how these commitments can be achieved. When evaluating access to NOE, researchers have measured: perceived distance [26,27,28]; objectively measured distance such as straight-line distance [29] or street-network distance [11]; amount of NOE close to home [30,31]; size of the nearest NOE [32,33,34]; or number of nearby NOE [30,31,35]. However, Schipperijn et al. [11] noted that the wide range of methods used to determine access to NOE made comparison between results difficult. Consequently, for example, evidence for a positive relation between NOE access and physical activity remains inconclusive. In an effort to create ‘harmonised methods’ for an urban NOE (specifically, green space) access indicator, Van den Bosch et al. [29] concluded their study with a recommendation of using a 300 m maximum straight-line distance to the boundary of urban NOE with minimum size of 1 hectare to determine which city residents had access to NOE. This study has informed the World Health Organization (WHO)’s distance and size recommendations, which has since been under discussion on whether there is sufficient evidence to support the recommended distance and size [36]. 

### 1.3. Aims of Study

This study aims to explore the relationship between access to NOE (combining distance to and size of NOE) and the actual use of NOE in different cities in Europe [11,29]. In order to do this, this study documented the relationship between residential access to NOE (using objective measures and including WHO distance and size recommendations) and use of NOE (self-reported time spent) using a large sample across four cities in Europe.

## 2. Materials and Methods

### 2.1. Study Setting

The data were obtained from the European Commission-funded PHENOTYPE study (Positive Health Effects of the Natural Outdoor environment in TYpical Populations in different regions in Europe) during which a cross-sectional survey was conducted (May to October 2013) in four cities across Europe that represented the range of setting in which most Europeans reside in terms of high-intermediate population density, size, climate and land cover: Stoke-on-Trent (United Kingdom); Doetinchem (The Netherlands); Barcelona (Spain); and Kaunas (Lithuania) (Nieuwenhuijsen et al. [37]).

Barcelona, with 15,968 inhabitants per km^2^, is a highly dense and urbanised city, with low greenery as indicated by the Normalized Difference Vegetation Index, NDVI, that in 2013 had a median (interquartile range, IQR) value of 0.21 (0.28). Doetinchem, on the contrary, is a highly green city (median (IQR)) NDVI in 2013 of 0.67 (0.54) with a low population density (706 inhabitants per km^2^). Stoke-on-Trent has a mean population density of 1194 per km^2^ and a median (IQR) NDVI in 2013 of 0.46 (0.15). Finally, Kaunas has a population density of 2047 per km^2^ and median (IQR) NDVI in 2011 of 0.64 (0.31). 

Natural outdoor environments (NOE) categories are defined in detail in Smith et al. [38], and size and percent cover are given in Table 1. Land use in and around each of the four cities is detailed in Smith et al. [38] and Kondo et al. [18].

### 2.2. Participants

Approximately 30 neighbourhoods per city which varied in socioeconomic status and typology, size and quantity of NOE were selected. For Barcelona, the neighbourhoods were defined based on census areas; for Doetinchem, postal codes; for Stoke-on-Trent, Lower-Level Super Output Areas (LSOAs) and for Kaunas, voting districts. As no comparable data existed for the four cities for socioeconomic status, each city used their own local data (i.e., local deprivation index in Barcelona and Stoke-on-Trent, household income in Doetinchem and education levels in Kaunas) to classify their neighbourhoods in tertiles. For NOE, Urban Atlas was used for all the cities where it was available (i.e., Barcelona, Stoke-on-Trent and Kaunas). For Doetinchem, Top10NL was used as this is a comparable dataset to Urban Atlas [38]. Based on these data, quintiles of neighbourhoods were done based on NOE availability. Tertiles of neighbourhood socioeconomic status and quintiles of neighbourhood NOE availability resulted in 15 different categories from which we selected our approximately 30 neighbourhoods (i.e., approximately two neighbourhoods per category). In each of these selected neighbourhoods, we conducted postal questionnaires (Kaunas) or administered face-to-face surveys (Barcelona, Doetinchem, Stoke-on-Trent). Approximately 1000 adults per city, aged 18–75 years, completed the survey. Accordingly, we obtained a total sample of n = 3947, as our overall response rate was 20%. Each participant provided written informed consent before taking part. The study was conducted in accordance with the Declaration of Helsinki. Ethical approval was obtained from the corresponding authority in each city: Clinical Research Ethics Committee of the Municipal Health Care (CEIC PS-MAR), Spain (2012/4978/I); Staffordshire University Faculty of Health Science ethics committee, United Kingdom; Medical Ethical Committee of the University Medical Centre Utrecht, the Netherlands; Lithuanian Bioethics Committee, Lithuania (2012-04-30 Nr.6B-12-147).

### 2.3. Measures

#### 2.3.1. Time Spent in Natural Outdoor Environments (NOE)

This outcome was measured by asking participants questions on frequency and duration of visits to NOE (Appendix A) and summed, according to the method used in previous studies [16]. 

#### 2.3.2. Objective Measures of Access to Natural Outdoor Environments (NOE)

Our objective assessment of access to NOE was based on residential addresses and it focused on two aspects: straight-line distance access and road network distance access to NOE. Accordingly, eight indicators of access to NOE were developed.

#### 2.3.3. Straight-Line Distance Access to Natural Outdoor Environments (NOE)

Straight-line access to NOE was estimated with the Urban Atlas land use map (Barcelona, Stoke-on-Trent, Kaunas) or Top10NL (Doetinchem). Both used a 1:10,000 scale and minimum represented unit of 0.25 ha (Top10 NL was adapted to be consistent with Urban Atlas). The included NOE categories were: Green Urban Areas, Agricultural and Semi Natural Areas, Forests. Straight-line distance to the nearest NOE was calculated using GIS. 

In brief, we estimated if participants’ residential addresses were at certain distances from NOE (details in Smith et al., [38]). Subsequently the presence of NOE at 150 m and 300 m was determined for NOE of at least 0.5 and 1.0 ha. We considered that an NOE was present if at least one NOE from the selected size was intersecting (i.e., overlapping) the participant’s residential straight-line distance buffer.

#### 2.3.4. Road Network Distance Access to Natural Outdoor Environments (NOE)

Road network access to NOE was estimated with the most detailed categorical and spatial land use maps we could get from local sources in each city. In brief, we used local road network maps to calculate if participants’ residential addresses were at certain distances from NOE (details in Smith et al. [38]). Land use categories were matched and adapted across cities to create comparable NOE categories. Finally, the NOE categories included to build these indicators were: parks, semi-natural urban spaces, natural/green corridors, formal recreation spaces (e.g., playgrounds), private gardens, amenity spaces (e.g., squares), street greenery, functional spaces (e.g., allotments), woodlands, derelict/vacant spaces, rural/agricultural land, country parks, lakes/reservoirs/ponds, rivers/streams/canals and coastal spaces.

Road network distance was chosen since it better reflects NOE accessibility as it takes into account the surrounding context and physical activity required to be physically present in an NOE (despite this, it is important to note that visual access linked to stress reduction and restoration may be equally and better represented by straight-line distance buffers, details in Smith et al. [38]). Residential road network distance to the nearest NOE was calculated using GIS. Subsequently, the presence of NOE at 150 m and 300 m was determined for NOE of at least 0.5 and 1.0 ha. We considered that an NOE was present if at least one NOE from the selected size was intersecting (i.e., overlapping) the participant’s residential network distance buffer. A geographical map showing the NOE distribution, road networks and residential location of participants is shown in Figure 1.

#### 2.3.5. Covariates

Different covariates were evaluated to be included for adjusting the model. Based in previous studies, dog ownership is associated with more time spent in NOE [39] and so was considered in this analysis as a covariate. Those variables that were related to the exposure and to the outcome with theoretical plausibility as potential confounding and kept significant after being included in the model at the same time were used at the final model (i.e., city, age, dog ownership (yes; no), childhood exposure to NOE, gender and educational level).

### 2.4. Analysis

Associations between distance to NOE and time spent in NOE were analysed, applying Poisson regression models to examine the associations between distance to NOE and time spent in NOE. Given this study’s cross-sectional nature, the coefficients are exponentiated regression coefficients (exp (β)), representing the proportional increase/decrease in time spent in NOE. 

The distribution of the exposure variables, access to NOE (distance and size), was analysed. Due to no normal distribution of the exposure variables, their range differences between cities and the nonlinearity of the effects, dichotomous exposure variables were created using meaningful cut-off points (i.e., presence vs. absence of the NOE of study at the explored distance). 

A total of 8 separated models for each dichotomous exposure variable were adjusted for the presence of NOE at 150 m and 300 m of at least 0.5 and 1.0 ha at straight-line distance and road residential network distance.

For the combined sample and for each city separately, crude and adjusted models were constructed with the latter adjusted for the city and the covariates outlined in Section 2.3.5. Goodness of fit of each model was evaluated to consider if the model was correctly fitted.

Associations with *p*-value ≤ 0.05 were considered statistically significant. All analyses were conducted using R version 3.6.1.

## 3. Results

Participants were on average 51.4 (SD = 11.4) years old and predominantly female (55.4%). The characteristics of the sample were statistically significantly different between the study cities (Table 1). For example, Kaunas had a higher median age and higher proportion of female respondents compared to other cities. These sample differences presented in Table 1 were controlled for in the analyses. For all our indicators of access to NOE, statistically significant differences between the cities were present. For these indicators, Barcelona was consistently the city with a lower proportion of participants having access to NOE (the only exceptions to this trend were the presence of NOE of 0.5 ha or more at 150 m and 300 m network distance from residences, indicators that had the lowest values in Kaunas). In contrast, Doetinchem was the city where a higher proportion of participants had access to NOE. No statistically significant differences were found in use of NOE between the four cities.

In general, we found that access to NOE was associated to use of NOE. Considering the results for the whole sample combined, we found that the stronger associations between use and access were for the 300 m straight line distance to NOE (both when considering a minimum size of 0.5 and of 1 ha). Considering the WHO recommendation, a significant positive association was found, with those living closer than 300 m spending 20% more time visiting NOE than those living further than 300 m away (exponentiated regression coefficient [expβ] = 1.20, 95% confidence interval [CI]: 1.19,1.22) (Table 2). The positive association was also found for the same distance to NOE of at least 0.5 ha (expβ = 1.21, 95% CI: 1.19,1.23). When the analysis was repeated for a straight-line distance of 150 m to NOE, the associations were slightly weaker.

The results from the Poisson regression models using street network distance also indicated that living closer to NOE is significantly associated with spending more time visiting NOE compared to those living further away, despite that the associations were weaker than those found with straight-line distances. For example, those living within less than 300 m of NOE of at least 1 ha spent only 5% more time visiting NOE than those living further than 300 m (exponentiated regression coefficient [expβ] = 1.05, 95% confidence interval [CI]: 1.04, 1.06). 

There were differences in associations between (and sometimes within) cities for each of the models (Table 2). Barcelona was the city that generally showed the strongest and most consistent results over all the access indicators. In Barcelona, living close to NOE (straight-line distance) was consistently associated with spending more than 30% more time visiting NOE than not living in the vicinity of an NOE. In Doetinchem, all the results were also in the direction of living closer to NOE being associated with more time spent in NOE, despite that some results were not statistically significant and most results showed a small relationship. In Stoke-on-Trent, the results were more mixed.

Living close to NOE (network distance) was consistently associated with spending at least 12% more time visiting NOE than not living close. Finally, the results for Kaunas were very mixed, showing positive associations for those living closer to 300 m (straight-line distance) to NOE of at least either 1 ha or 0.5 ha and for those living at less than 150 m (road network distance) to NOE of at least 1 ha, but inverse (negative) associations for those within 300 m (road network distance) to NOE of at least either 1 ha or 0.5 h. The only model where all cities separately and combined showed a significant positive association with time spent in NOE was for those living within 150 m network buffer of 1 ha NOE.

## 4. Discussion

This study explored the relationship between access and use of natural outdoor environments in four European cities. The results showed that all NOE access indicators were positively associated with higher use of NOE with the pooled dataset, despite that the strongest associations were for the 300 m straight distance to NOE of at least 0.5 ha and the WHO recommendation for the same distance to NOE of at least 1 ha. 

However, between- and within-city differences existed. Barcelona was the city that showed the strongest and most consistent results over all the access indicators. In Doetinchem, all the results were also in the direction of the pooled results (despite some results not being statistically significant). In Stoke-on-Trent, and especially in Kaunas, the results were more mixed. The only model where all cities separately and combined showed a significant positive association with time spent in NOE was for those living within 150 m network distance of 1 ha NOE.

Our findings for 150 m straight distance, 150 m network distance, 300 m straight distance and 300 m network distance correspond with previous research using objectively measured time spent accessing NOE [20,40,41]. However, our findings of residential 150 m network buffer access to NOE of at least 1 ha are more consistently associated with the use of NOE than the WHO recommendation are novel. Our results may indicate the importance of the size of publicly accessible NOE, which should be of 1 ha minimum, in agreement with previous research [29,36,42]. 

Our results were not consistent across all cities. Barcelona is the densest city in our sample with very little NOE but high walkability, so residents may make use of any available NOE regardless of size or distance. In contrast, Doetinchem is very green with 80% or more of participants having access to NOE across the different NOE indicators we considered, meaning residents could be more selective in choosing NOE available between 150 m and 300 m. Stoke-on-Trent has varying access to NOE over the different indicators and is one of the less walkable cities from our study sample, placing reliance on nearby NOE. Kaunas also has varying access to NOE and lower walkability, but its results conflict with the combined sample. Kaunas’s results could indicate that other variables (such as neighbourhood safety or quality of NOE) may have an important role in determining the use of NOE in this city. In summary, the differences we found by city could indicate the importance of other factors in the relationship between distance and use of NOE such as the land use, life organisation and other factors that may affect the experience and tendency of residents to visit NOE and may determine what role residential NOE has (in comparison to workplace greenspace, for example) and if people do not compensate a lack of NOE close to residence with visits to NOE that are further from the home or not. 

### 4.1. Study Scope

It is not within the scope of this article to explore the associated benefits of time spent within NOE, and we do not claim that the increase in time spent in NOE infers any associated benefits. For example, viewing NOE without physically visiting has been shown to offer a range of health benefits (both perceived and objectively measured) [43,44,45]. However, opportunities and rights to view NOE are not usually considered in planning matters, whereas distance is broadly accepted as a material consideration in planning when linked with policy. 

Despite not being part of the scope of the present study, differences by population subgroup on the association between access and visits to NOE (and their potential health benefits) may exist. Contrasting associations may arise due to complex socio-natures and past experiences, differential vulnerability to other city processes such as gentrification and its associated physical and sociocultural displacement threats, structural inequalities and power imbalances (or even exclusion) of certain groups from input and decision-making about planning, design and management of NOE [46,47]. However, while consideration of measures such as use, social contact and geographical and social contexts are important in research, these are also difficult to apply in urban planning policy. 

### 4.2. Policy Implications 

According to our findings on the use of NOE, it is advisable to reduce the World Health Organization’s recommended distance from 300 m to 150 m network distance to 1 ha of NOE if an indicator for—at least—the whole of Europe is the aim. This is likely to lead to people spending more time in NOE and reaping the associated public health benefits associated with exposure to NOE, which have been outlined in the introduction of this paper. Although the findings indicate that exploration of network buffers would be advisable, straight-line distances may be easier to document in policy and to apply in practice for planning practitioners as they can identify and agree upon requirements for developers and planning bodies. For research purposes, a street network buffer of 150 m should be considered wherever possible. The recommended minimum size of 1 ha of publicly accessible NOE is further supported by this research. 

### 4.3. Limitations

As is to be expected with research of this magnitude, our study faced some limitations. Our cross-sectional study had a limited capacity to infer causality in the evaluated associations, and we were also unable to rule out self-selection bias. As such, people more inclined to use NOE frequently could purposely live within areas with higher NOE, rather than incurring higher use from higher residential NOE availability. Moreover, our results are susceptible to measurement error as our objectively measured residential distances to NOE do not consider important aspects such as access points or quality of NOE, and our use of NOE measures are self-reported. Further to this, self-reported time spent in nature does not necessarily translate to exposure to nature. Last, whilst our use indicators included both green and blue space, our access indicators only included green space. We explored this potential limitation by creating an indicator adding the comparable green and blue spaces variables for the 150 m and 300 m cut-offs and found no substantial differences with the indicators we used in our models.

Samples were not equally distributed between the four cities and differences in demographics were observed; therefore, estimates may have residual confounding by unknown factors that could vary between study areas [5]. Moreover, whilst our focus considered access to the nearest NOE, there may be a number of factors which influence choice of NOE, and the closest space may not be used most by people. However, the protocol we followed in all cities was consistent, and our approach was based in methodology widely used in previous research [16,37,38]. An important strength of our study is that data were collected in four different European cities, enabling us to compare results across cities with regional, social and cultural differences. Moreover, our study is one of the first to test current guidance on distances to NOE across several European cites using the measure as the number of hours of purposeful visits to NOE, which was used to inform the WHO guidance.

## 5. Conclusions

This study contributes to the discussion of embedding recommended distances to NOE in planning policy and provides further evidence for providing greater opportunity for access to NOE. Data on use of and access to NOE from our large sample of adults across four different European cities indicated that the NOE indicator more strongly associated with time spent in natural environments was the straight-line 300 m indicator (either the WHO recommendation based on NOE of 1 ha or the indicator based on 0.5 ha spaces). However, the only indicator that was consistent across all cities was living within a 150 m network buffer of NOE (of at least 1 hectare). Therefore, it is advisable to explore the possibility of reducing the World Health Organization’s recommended distance to NOE to 150 m network distance to 1 ha of NOE for a general indicator for the whole of Europe. Our results further support the WHO recommendation that the size of publicly accessible NOE should be a minimum size of 1 ha.

## Figures and Tables

**Figure 1 ijerph-19-02226-f001:**
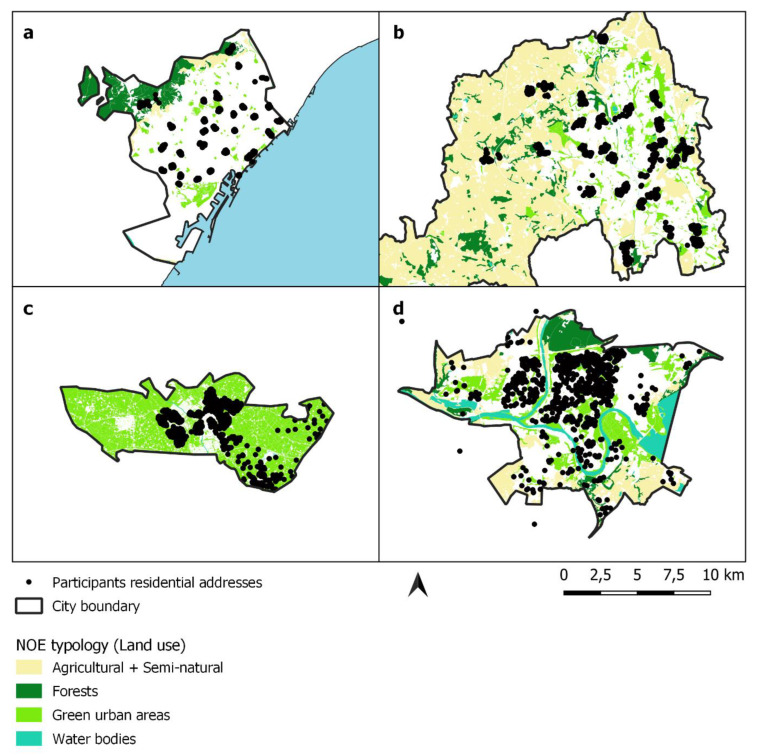
A geographical map showing the NOE distribution, road networks and residential location of participants. Map showing the four study areas and the distribution of the NOE features from Urban Atlas or the ‘Top 10 NL’ for Doetinchem. (**a**) Barcelona (**b**) Stoke-on-Trent (**c**) Doetinchem and (**d**) Kaunas.

**Table 1 ijerph-19-02226-t001:** Sample characteristics and differences.

	Total	Barcelona	Stoke-on-Trent	Doetinchem	Kaunas	
*Sociodemographic characteristics*						
*Gender*, females (n (%))	2185 (55.4)	551 (52.8)	542 (51.9)	488 (56.7)	604 (60.6)	*
*Age* (years: median (IQR))	51.4 (11.4)	45 (15.5)	45.8(16.1)	56.4 (12.2)	59.5 (14)	*
*Education*						*
Low (n (%))	274 (6.9)	150 (14.4)	97 (9.5)	10 (1.2)	17 (1.7)	
Medium (n (%))	1732 (43.9)	403 (38.7)	657 (64.3)	408 (47.4)	263 (26.4)	
High (n (%))	1915 (48.5)	488 (46.9)	268 (26.2)	442 (51.4)	717 (71.9)	
*Owning a dog*, yes (n (%))	1253 (31.7)	201 (19.3)	336 (32.2)	184 (21.4)	532 (53.4)	*
*Childhood NOE experience*						*
Never (n (%))	97 (2.5)	36 (3.5)	33 (3.2)	7 (0.8)	21 (2.1)	
Sometimes (n (%))	517 (13.1)	161 (15.5)	214 (20.8)	55 (6.4)	86 (8.6)	
Regularly (n (%))	902 (22.9)	173 (16.6)	259 (25.1)	139 (16.1)	331 (33.2)	
Often (n (%))	954 (24.2)	268 (25.7)	195 (18.9)	244 (28.3)	247 (24.8)	
Very often (n (%))	1461 (37)	403 (38.7)	330 (32)	416 (48.3)	312 (31.3)	
*Time in NOE* (hours/month: median (IQR))	49.12 (77)	49.12 (80.25)	47.38 (113.63)	60.50 (42.88)	47.88 (84.50)	
*Access to NOE (distance and size)*						
*Presence of NOE of 0.5 ha or more at 150 m straight-line distance* (n (%))	1629 (41.3)	498 (47.7)	708 (67.8)	762 (88.5)	615 (61.7)	*
*Presence of NOE of 1 ha or more at 150 m straight-line distance* (n (%))	1816 (46)	229 (21.9)	654 (62.6)	677 (78.6)	571 (57.3)	*
*Presence of NOE of 0.5 ha or more at 300 m straight-line distance* (n (%))	737 (18.7)	498 (47.7)	987 (94.5)	861 (100)	864 (86.7)	*
*Presence of NOE of 1 ha or more at 300 m straight-line distance* (n (%))	854 (21.6)	453 (43.4)	975 (93.4)	832 (96.6)	833 (83.6)	*
*Presence of NOE of 0.5 ha or more at 150 m network distance* (n (%))	1993 (50.5)	416 (39.8)	491 (47)	700 (81.3)	347 (34.8)	*
*Presence of NOE of 1 ha or more at 150 m network distance* (n (%))	2361 (59.8)	308 (29.5)	436 (41.8)	516 (59.9)	326 (32.7)	*
*Presence of NOE of 0.5 ha or more at 300 m network distance* (n (%))	872 (22.1)	800 (76.6)	857 (82.1)	846 (98.3)	572 (57.4)	*
*Presence of NOE of 1 ha or more at 300 m network distance* (n (%))	1302 (33)	560 (53.6)	802 (76.8)	743 (86.3)	540 (54.2)	*

n, number of participants; IQR, interquartile range; NOE, natural outdoor environment. * statistically significant difference (*p* < 0.001) between cities at 5% alpha level based on chi-square and Kruskal−Wallis tests.

**Table 2 ijerph-19-02226-t002:** Associations between access to NOE (distance and size) and use of NOE (time spent in NOE) in the combined sample and by city.

	Combined Sample	Barcelona	Stoke-on-Trent	Doetinchem	Kaunas
Presence of NOE of...	Expβ (95% CI)	Expβ (95% CI)	Expβ (95% CI)	Expβ (95% CI)	Expβ (95% CI)
0.5 ha or more at 150 m straight-line	1.10 (1.09, 1.11) ***	1.34 (1.31, 1.37) ***	1.16 (1.13, 1.18) ***	1.00 (0.97, 1.03)	0.98 (0.97, 1.00)
1 ha or more at 150 m straight-line	1.12 (1.10, 1.13) ***	1.36 (1.33, 1.39) ***	1.21 (1.18, 1.24) ***	1.04 (1.02, 1.07) ***	0.99 (0.98, 1.01)
0.5 ha or more at 300 m straight-line	1.21 (1.19,1.23) ***	1.31(1.29,1.34) ***	0.82 (0.78,0.85) ***	#	1.14 (1.11,1.17) ***
1 ha or more at 300 m straight-line	1.20 (1.19, 1.22) ***	1.41 (1.38, 1.44) ***	0.81 (0.78, 0.85) ***	1.03 (0.97, 1.10)	1.08 (1.05, 1.11) ***
0.5 ha or more at 150 m network	1.08 (1.07, 1.09) ***	1.13 (1.10, 1.15) ***	1.26 (1.23, 1.29) ***	1.02 (1.00, 1.05)	0.99 (0.97, 1.01)
1 ha or more at 150 m network	1.12 (1.11, 1.13) ***	1.21 (1.19, 1.24) ***	1.34 (1.32, 1.37) ***	1.03 (1.01, 1.05) ***	1.04 (1.02, 1.06) ***
0.5 ha or more at 300 m network	1.02 (1.00, 1.03) *	1.00 (0.98, 1.03)	1.17 (1.14, 1.20) ***	1.52 (1.38, 1.67) ***	0.95 (0.94, 0.97) ***
1 ha or more at 300 m network	1.05 (1.04, 1.06) ***	1.12 (1.10, 1.15) ***	1.12 (1.09, 1.15) ***	1.09 (1.05, 1.12) ***	0.96 (0.95, 0.98) ***

Based on Poisson model analyses adjusted for age, dog ownership, childhood nature exposure, gender and educational level (and city for the combined sample analysis). NOE = natural outdoor environments; expβ = exponentiated regression coefficient; CI = confidence interval; ref = reference group. # No estimable model for Doetinchem due to not having observations in the category of absence of NOE of 0.5 ha or more at 300 m straight-linear distance for the exposure variable. * *p* ≤ 0.05; *** *p <* 0.001.

## Data Availability

The data presented in this study are available on request from the corresponding author. The data are not publicly available due to privacy and ethical concerns.

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
