# Peer review of "Use of the Natural Outdoor Environment in Different Populations in Europe in Relation to Access: Implications for Policy"

_ijerph, 2022, doi:10.3390/ijerph19042226_

Round 1

Reviewer 1 Report

This is an interesting and informative, well-conducted study about the use and access to the natural outdoor spaces (NOE), where authors conclude that the policy recommendation for situating these spaces should be 150m, and not 300m in the straight line from the residence, as recommended before by WHO. This is to ensure the more frequent use of the green spaces by the public. The use was estimated with the "time spent" self-reports, this however doesn't directly translate to the exposure to nature, and specific types of nature, which is linked to the improved mental health and well-being. This should be discussed in the limitations section.

Some minor comments:

Just a suggestion about the title - it currently doesn't seem to convey the main strong message to the reader, which is actually quite ground-breaking. Suggest to reformulate it.

[line 24] - "Exploring this.." what exactly this refers to? This leads to the research question lacking. Do authors mean to check whether current recommendations are valid?

[line 49] - do authors mean reduced stress?

Authors to explain how WHO arrived at the 300m recommendation. 

Authors to consider adding the emotion regulation/mood improvement/reduction of depression in the list of benefits from NOE visit, and suggested literature:

Olszewska-Guizzo, A., Sia, A., Fogel, A., & Ho, R. (2020). Can exposure to certain urban green spaces trigger frontal alpha asymmetry in the brain?—Preliminary findings from a passive task EEG study. International journal of environmental research and public health17(2), 394.

Bratman, G. N., Olvera‐Alvarez, H. A., & Gross, J. J. (2021). The affective benefits of nature exposure. Social and Personality Psychology Compass.

[line 61] I believe there are more recent reference supporting this statement, for example: Remme, R. P., Frumkin, H., Guerry, A. D., King, A. C., Mandle, L., Sarabu, C., ... & Daily, G. C. (2021). An ecosystem service perspective on urban nature, physical activity, and health. Proceedings of the National Academy of Sciences118(22).

[line 328] "generally safe European cities like Barcelona, Paris or London", please provide evidence for that in light of an example.  Why safety is a factor in WHO recommendation?

Section 4.3-  Authors suggest to reduce the WHO recommendation from 300 to 150m, but they should be transparent about the target that this will achieve. According to their findings, this recommendation will likely increase the use of NOE, the influence on WHO goals (better public health measures) is not explored by this study.

Author Response

Thank you for taking time to review our manuscript and for your positive and constructive feedback. Please see the attachment.

Reviewer 2 Report

First suggestion is to write out NOE, as most readers will not understand what it refers to. The abbreviation can be used selectively within a paragraph or two, then written out again. Otherwise it makes things difficult for the reader. 

Second, the basic and classic references are missing. Alexander et al.'s measure of access to local green space lies behind the authors' results. It is "Accessible Green", Pattern No. 60. 

https://en.wikipedia.org/wiki/A_Pattern_Language

If the authors don't have access to this classic book, the pattern is described here:

https://ojs.emu.edu.tr/index.php/jurd/article/view/240/128

Continuing with the pattern concept, there are relevant new patterns for urban green space access here, especially "Neighborhood Park", Pattern 4.4:

https://pattern-language.wiki/.../Table_of_Contents_%28NPL%29

One omission is not to mention why green space access is so important to human health, which is what this manuscript measures. The answer is Biophilic Urbanism, included as Pattern No. 2.4 in the New Pattern Language. This is worth explaining to support the motivation for the research. 

The Conclusion is correct, but tentatively stated. As this is a serious public health issue, the authors could be somewhat more insistent on changing present-day regulations to encourage more embedded green space in cities, in a better distribution than is found today. 

Finally, this topic has been extensively studied, and I notice that several recent references are missing. The authors may find these in the following articles:

https://usfblogs.usfca.edu/sustainability/2020/05/19/the-values-and-shortcomings-of-green-spaces-in-urban-environments/

https://ncceh.ca/sites/default/files/Full_Review-Greenspace_Mental_Health_Mar_2015.pdf

https://www.euro.who.int/__data/assets/pdf_file/0010/337690/FULL-REPORT-for-LLP.pdf

I'm sure that a more inclusive list of references will improve this manuscript. 

Author Response

Thank you for taking time to review our manuscript and for your constructive feedback. Please see the attachment.

Reviewer 3 Report

The authors should state the details about the methods this study used  in Methods section, instead of using a simple description of "Poisson regression models". How many models did the authors applied? What was the formula? What was the explained variable and what were the explanatory  variables? Was the model used in the combined sample identical to those used in individual cities? In brief, the authors must rewrite the method part to make sure the readers can clearly understand the corresponding methods.

Moreover, the authors reported there were statistically significant differences or not between four cities  in main text (also in Table 1). But what methods did the authors use for this test? I haven't seen anything in the paper. Please add the statement about the test  in Methods section. In addition, what is the meaning of "P value" in the last column of Table 1?

The use of NOE includes two issues: "how often" and "how much time" . But why did the authors just list "time in NOE" in Table 1? And what is the meaning of the numbers in this item (e.g., 49.12 hours)? Please clarify the calculation of the "time spent in NOE" variable.

The layout of Table 1 is unfriendly for the readers. The authors should reorganize the table to make clear the descriptive statistics of the raw data.

Again, without the clear statement about the methods, Table 2 and the corresponding part in main text are meaningless for the readers.

The results in this study seem to have weak support to the policy implications. The findings are not convincing enough.

I have no idea why the authors kept some references to be masked (e.g., lines 158, 172, 191)? Haven't they been published yet?

At last, I strongly recommend the authors add the geographical maps to show the NOE distributions in four cities, the road networks, and the residential locations of participants. I also suggest the authors report the questionnaire design of the survey as supplemental materials.

Author Response

(The authors gave the same response as above.)

Round 2

Reviewer 1 Report

Authors have improved the ms according to suggestions.